# Timing of Resection of Spinal Meningiomas and Its Influence on Quality of Life and Treatment

**DOI:** 10.3390/cancers16132336

**Published:** 2024-06-26

**Authors:** Michael Schwake, Wesam Said, Marco Gallus, Emanuele Maragno, Stephanie Schipmann, Dorothee Spille, Walter Stummer, Benjamin Brokinkel

**Affiliations:** 1Department of Neurosurgery, University Hospital Münster, 48149 Münster, Germany; wesam.said@ukmuenster.de (W.S.); marco.gallus@ukmuenster.de (M.G.); emanuele.maragno@ukmuenster.de (E.M.); stephanie.schipmann@ukmuenster.de (S.S.); dorotheecaecilia.spille@ukmuenster.de (D.S.); walter.stummer@ukmuenster.de (W.S.); benjamin.brokinkel@ukmuenster.de (B.B.); 2Department of Neurosurgery, University of Bergen, NO-5020 Bergen, Norway; 3Department of Neurosurgery, Clemenshospital, 48153 Münster, Germany

**Keywords:** spinal meningioma, spinal tumor, quality of life, quality of care

## Abstract

**Simple Summary:**

Is the resection of spinal meningiomas in asymptomatic patients or patients with mild neurological symptoms justified? In this study, we compare the neurological outcome, quality of life, and quality of care of these patients to patients with more severe neurological symptoms. The results show that early neurosurgical intervention leads to better neurological outcomes and quality of life, contradicting a watch-and-see regime.

**Abstract:**

Background: The main treatment modality for spinal meningiomas (SM) is gross total resection (GTR). However, the optimal timing of surgery, especially in cases with absent or mild neurological symptoms, remains unclear. The aim of this study is to assess the impact of early-stage resection on neurological outcome, quality of life (QoL), and quality of care. The primary objective is a favorable neurological outcome (McCormick scale 1). Methods: We retrospectively analyzed data from patients who underwent operations for SM between 2011 and 2021. Patients with mild neurological symptoms preoperatively (McCormick scale 1 and 2) were compared to those with more severe neurological symptoms (McCormick scale 3–5). Disabilities and QoL were assessed according to validated questionnaires (SF-36, ODI, NDI). Results: Age, spinal cord edema, thoracic localization, and spinal canal occupancy ratio were associated with more severe neurological symptoms (all *p* < 0.05). Patients presenting with mild symptoms were associated with favorable neurological outcomes (OR: 14.778 (95%CI 3.918–55.746, *p* < 0.001)), which is associated with shorter hospitalization, better QoL, and fewer disabilities (*p* < 0.05). Quality of care was comparable in both cohorts. Conclusions: Early surgical intervention for SM, before the development of severe neurological deficits, should be considered as it is associated with a favorable neurological outcome and quality of life.

## 1. Introduction

Spinal meningiomas (SM) are benign, intradural, juxtamedullary tumors situated within the spinal canal that may cause neurological deficits and pain contingent upon their localization. The primary objective of the treatment is to achieve gross total resection (GTR) to allow neurological recovery and mitigate the risk of tumor recurrence [1,2,3,4]. Due to the benign nature of these tumors, the neurological deficits manifest slowly. A subset of patients may exhibit incidental radiological findings indicating the presence of SM with no or minor neurological deficits.

There is a growing focus on quality of life (QoL) and quality of the delivered care [5]. Quality indicators (QI), predominantly proposed and implemented by healthcare policymakers, such as readmission and reoperation rates evaluating treatment and care, are continually evolving and have been utilized for reimbursement purposes [6,7]. Beyond that, for the patients, QoL is a critical factor, as it significantly impacts their ability to resume social and employment activities.

In the past, several publications showed relatively good outcomes and QoL after resection of intra-spinal, intradural tumors, including meningiomas. However, most of these publications reported bilateral laminectomy as the main approach for tumor resection [4,8,9,10]. Recently, we demonstrated that the resection of spinal meningiomas is feasible via a less invasive unilateral approach, showing a similar rate of gross total resection (GTR) in comparison to a bilateral laminectomy. Furthermore, we demonstrated that patients undergoing unilateral hemilaminectomy, with less muscle detachment and bone resection, had significantly less blood loss (EBL) during the surgical procedure and faster recovery with significantly shorter length of hospital stay (LOS) [11]. Minimal-invasive spine surgery (MISS) can further improve surgical outcomes through reduced surgical impact, decreased pain, and quicker recovery [12,13,14]. The European guidelines on diagnosis and management of meningiomas recommend GTR as first-line treatment; however, the decision to offer surgery rather than observation should balance the benefit of tumor removal versus surgical risk [15].

The aim of this study is to question the optimal timing of resection of spinal meningiomas and its influence on QoL and QI. For these reasons, we compared patients who were operated on in an early stage of the disease with mild symptoms (McCormick scale 1–2) and patients who were operated on with more severe symptoms (McCormick scale >2). The primary outcome was favorable neurological recovery (McCormick scale 1), and whether performing surgery on patients with mild neurological symptoms is recommended. The secondary outcomes are QI, including length of hospital stay, nosocomial infections, 90 days unplanned readmission, and re-surgery. In addition, we evaluated QoL and postoperative disability according to short form 36 (SF-36), Oswestry disability index (ODI), and neck disability index (NDI) questionnaires.

## 2. Materials and Methods

### 2.1. Study Design

We included all patients who underwent resection of spinal meningioma between 2011 and 2021 at our neurosurgical department in this retrospective study. The following data from the hospital electronic records were analyzed: age, sex, tumor volume, spinal canal occupancy ratio, neurological symptoms, localization of the tumor within the spinal canal and its relation to the spinal cord, the presence of cord edema, surgical approach, the extent of resection (EOR) according to the Simpson classification [16], use of intraoperative neuromonitoring (IOM), estimated blood loss (EBL; mL), duration of surgery (minutes), length of hospital stay (LOS; days), 90-day nosocomial infections, 90-day surgical site infection, unplanned 90-day readmission and 90-day reoperation, 90-day mortality, and tumor progress or recurrence.

Furthermore, we contacted the patients and sent them questionnaires, including an SF-36 questionnaire to evaluate QoL. To evaluate their functional outcome, we used the Oswestry disability index (ODI) and neck disability index (NDI) depending on the localization of the tumor within the spinal canal [17,18]. Back, neck, or extremity pain was determined according to a visual analog scale (VAS) from 0 to 10. Prior to sending the questionnaires, patients were contacted by telephone and informed about the study. Informed consent was obtained from each patient participating in this part of the study.

The study was conducted in accordance with the Declaration of Helsinki and approved by the Institutional Review Board of the University of Münster, Germany (reference number 2021-714-f-S, 15 February 2022).

### 2.2. Surgical Intervention

Surgical resection was performed via a minimal-invasive dorsal or dorso-lateral approach using hemilaminectomy according to the tumor’s location in the spinal cord. During surgery, intra-operative neurophysiological monitoring (IOM, inomed Medizintechnik GmbH, Emmendingen, Germany) was conducted, including motor-evoked potentials (MEP) and sensory-evoked potentials (SEP) of upper and lower extremities. After the exposure of the dura, intraoperative sonography was performed to detect the tumor, and the dura was opened. After the durotomy, the nerve roots were identified, and in ventrally located tumors, the dentate ligaments were also identified and cut. Subsequently, the tumor poles were visualized. In order to be able to remove the tumor through the laminotomy, it was debulked using an ultrasonic aspirator (CUSA^®^, Integra Lifesciences, Princeton, NJ, USA). Finally, the meningioma was removed, and the dura attachments were coagulated to achieve resection grade 2 according to the Simpson classification whenever feasible [16]. The dura was closed using a 6-0 monofil continuous suture.

### 2.3. Outcome and Assessment of QoL and Functionality

The neurological status of the patients was assessed three months after surgery using the modified McCormick scale [19], ranging from 1 (no symptoms or minimal dysesthesia) to 5 (paraplegic/quadriplegic). The status was evaluated both before and after surgery independently by two of the authors (MS and WS). For further dichotomic calculations, the McCormick scales 1 and 2 were considered as ’mild symptoms’, and 3 to 5 as ‘severe symptoms’. Postoperatively, a McCormick scale of 1 was considered a ‘favorable recovery’.

To assess QoL and functional disabilities, patients were subsequently contacted and requested to complete standardized questionnaires. QoL was assessed by the SF-36 questionnaire. It contains questions evaluating general health, physical functioning, limitations due to physical health, limitations due to emotional problems, energy/fatigue, emotional well-being, social functioning, pain, and health change. We compared the results of the patient population with the general population from data published previously [20].

Postoperative disability was measured with ODI or NDI questionnaires. Patients with meningiomas in the thoracic and lumbar spine answered the ODI questionnaire, and those with tumors in the cervical spine the NDI questionnaire. In addition, all patients were asked to evaluate back, neck, or extremity pain using a VAS score of 0 to 10.

### 2.4. Evaluation of Images

Tumor volume was semiautomatically measured using Brainlab Elements^®^ software (Brainlab AG, Munich, Germany) and expressed in milliliters (mL). Furthermore, we calculated the occupancy ratio of the meningioma in comparison to the spinal canal by measuring the area of the meningioma on the slide with the largest extension and dividing it by the area of the spinal canal; the ratio is shown in percentage [21]. In addition, we evaluated whether the meningioma was localized ventrally of the dentate ligament or posterior to it and whether spinal cord edema was present or not.

The first postoperative MRI was performed three months after surgery. Afterward, imaging was repeated after one year; subsequently, surveillance intervals were doubled after each unremarkable MRI.

### 2.5. Statistical Analysis

Statistical analysis was performed using SPSS Statistics 29.0 (IBM Corp., Armonk, NY, USA). Categorical variables are shown as absolute and relative frequencies. Parametric values are presented in mean and standard deviation (SD). Non-parametric values are presented as the median and interquartile range (IQR, 25% quartile and 75% quartile). Fisher’s exact test was performed to compare groups of binary categorical variables. A two-tailed Student’s *t*-test was used as a parametric, and a two-sided Mann–Whitney U-test (MWU) as a non-parametric test. A probability value less than *p* < 0.05 was considered statistically significant.

## 3. Results

### 3.1. Patients’ Characteristics

In the 10 years between 2011 and 2021, 65 cases of spinal meningiomas were operated at our department. Female patients accounted for the vast majority, with 59 cases (90.77%). Two of the patients underwent surgery due to tumor recurrence (*n* = 2, 3.08%), and one due to progression after partial resection (*n* = 1, 1.54%). Patients’ ages at the time of surgery ranged between 25 and 86 years, with a mean age of 58.4 (±14.10). In most of the cases, the meningiomas were located in the thoracic spine (*n* = 45, 69.23%), followed by 17 (26.15%) cases in the cervical spine, and only rarely in the lumbar spine (*n* = 3, 4.62%). We noticed more cases at junction areas C1, T1 to 4, and T 8 to 12 (Figure 1). Mean tumor volume was 1.22 mL (±0.85), and mean occupancy ratio of the spinal canal was 51.88% (±19.85%). Cord compression was visible in 91% of the cases (*n* = 59), in 20% of the cases, cord edema was present (*n* = 13), and all patients included in this study had a grade I tumor according to the WHO classification (*n* = 65, 100%). Interestingly, in seven cases (10.77%), patients had a history of cranial meningioma surgery, too.

For further analysis, we divided the patients into two cohorts. The first consisted of patients with mild symptoms (McCormick scale 1 and 2), and the second patients with more severe symptoms (McCormick 3 and higher). In this analysis, we found out that patients with mild symptoms were younger (*p* = 0.015). Preoperatively, they had better Karnofsky performance scale, less gait ataxia, less motor weakness, and fewer sensory deficits (all *p* < 0.001) compared to patients with more severe symptoms. In addition, patients with mild symptoms had a higher percentage of radicular and local pain (*n* = 18, 42.86% and *n* = 16, 38.10%, respectively, in comparison to *n* = 6, 26.09% and *n* = 5, 21.74%), however, not reaching statistical significance (both *p* > 0.05).

Although the tumor volume was comparable in both cohorts, the spinal canal occupancy ratio—measured at the level of the tumor’s largest diameter—was higher in the cohort with more severe neurological symptoms (*p* = 0.016). In addition, patients with severe symptoms more frequently had meningiomas in the thoracic spine (*p* = 0.026) and spinal cord edema (*p* = 0.049). See Table 1 and Figure 2 for further information.

### 3.2. Neurological Outcome and Quality Indicators

We noticed an improvement in at least one of the neurological symptoms and pain levels in most cases (*n* = 64, 98.46%). However, patients with mild symptoms had higher odds for favorable outcomes (postoperative McCormick scale 1; 14.7778 (95%CI 3.9175–55.746, *p* < 0.001)). The least improvement was noticed in bladder function in both cohorts. Furthermore, the recovery of patients with mild symptoms preoperatively was faster, and their LOS was significantly lower, with a mean LOS of 7.07 days (±2.4) in comparison to 10.04 days (±5.36, *p* = 0.003). While the Karnofsky performance scale improved in both cohorts, it was still significantly better in the cohort of patients with mild symptoms (*p* = 0.004). Although improved in both cohorts, sensory loss was more frequent in the cohort with severe symptoms (*p* = 0.006). All other symptoms and deficits were comparable.

Adverse events were very rare in both cohorts, and we noticed only one nosocomial infection, one unplanned readmission, and one re-surgery due to CSF leakage within 90 days; see Table 2 and Figure 3 and Figure 4 for further information.

### 3.3. Functionality and Quality of Life

Out of the patients we were able to contact, we received 38 (60% of all patients) completed questionnaires. Six of the contacted patients did not return the questionnaire. Two sent inadequately filled forms, one patient had passed away, and sixteen patients could not be reached. Two of these patients who replied had undergone secondary surgery due to tumor recurrence. The mean time between surgery and contact was 6 years (±3.69). Overall, patients reported good functionality (ODI/NDI 0–20%) in 28 (73.68%) cases. Six patients (15.79%) had moderate disabilities (ODI/NDI 21 to 40%), while four (10.53%) patients had more severe disabilities (ODI/NDI 41 to 62%). The four patients with severe disabilities were relatively old, with a mean age of 76 (range 68 to 83). Remarkably, the three younger patients (age range 68–78) had severe preoperative neurological deficits, all classified as McCormick grade 4. This suggests that a poor neurological status in combination with advanced age played a major role in their functional disabilities.

The results of the SF-36 questionnaire revealed that the QoL of patients after resection of spinal meningiomas is quite similar to the general population when compared to data published previously [20]. One exception is the slightly reduced physical function. Patients included in this study reported a mean of 71.32 in this part of the questionnaire in comparison to 83.7 in the general population. On the other hand, patients after resection of spinal meningiomas had slightly fewer complaints regarding pain, see Figure 5. One important piece of additional information is that in the study on QoL in the general population, the mean age was notably lower than the mean age in this study. The physical function score had a strong negative correlation to the ODI/NDI values (r(36) = −0.83, *p* < 0.001), see Figure 6.

To compare functionality and QoL and neurological outcomes, we allocated between patients with favorable neurological outcomes and those with incomplete neurological outcomes. This analysis revealed a significantly better disability index in patients with favorable neurological outcomes. The patients with favorable neurological outcomes had a mean ODI/NDI score of 9.26% (±11.60%) in comparison to 23.27% (±24.10, *p* = 0.020) in the cohort of patients with incomplete neurological outcomes. This was in concordance with the SF-36 subcategories of physical function and role limitations due to physical health (both *p* < 0.05). All other subcategories showed no significant differences.

As described above, the main risk factor for incomplete neurological outcome was the preoperative neurological status. Patients with preoperative McCormick scale 1 and 2 had significantly more favorable outcomes than those with preoperative McCormick scale 3 and 4 (*p* < 0.001). In addition, the neurological outcome was associated with LOS, with a mean of 6.85 days (±1.2). The LOS was significantly lower in patients with favorable neurological outcomes compared to 9 days (±4.2) in cases with incomplete neurological outcomes (*p* = 0.022). See Table 3, Table 4 and Table 5 for further information.

### 3.4. Tumor Recurrence and Progression

Three patients (4.6%) had a tumor recurrence after a mean follow-up period of 7.12 (±3.95) years. One of these patients had a resection grade 3 according to the Simpson classification, showing some tendency, however, without statistical significance (*p* = 0.196). On the other hand, two out of the three patients with a resection grade 3 without progression had postoperative stereotactic irradiation.

## 4. Discussion

The neurological and functional outcomes following resection of spinal meningiomas are excellent in most cases. However, patients with mild preoperative symptoms experience more favorable recoveries compared to those with severe preoperative symptoms. A favorable neurological outcome is associated with better disability scores and improved quality of life (QoL). The findings of this study suggest that performing tumor resection in patients with mild neurological symptoms is justified to achieve optimal neurological outcomes, functionality, and quality of life.

### 4.1. Neurological Outcome after Minimal-Invasive Resection

Our results indicate that the postoperative status following resection of spinal meningiomas depends mainly on the preoperative neurological status of the patients; in addition, the age of the patients seems to play a role [2,8]. Patients with no or mild symptoms, who were classified preoperatively as McCormick scale 1 or 2 experienced favorable neurological outcomes in most cases. On the contrary, patients with more severe neurological symptoms experienced improvement of their deficits in most cases; however, significantly fewer patients had favorable neurological outcomes, reaching McCormick scale 1 after surgery in comparison to patients with mild symptoms preoperatively. Nonetheless, in comparison to intramedullary lesions such as hemangioblastoma, the recovery rate was much better [22]. These results are in line with previously published studies investigating outcomes after other intradural tumor resections [2,21,23] and highlight that severe neuronal damage would not fully recover after surgery.

Patients with mild symptoms in this study had more frequent pain as a primary symptom, in comparison to more severe neurological deficits such as gait ataxia, sensory deficits, or motor weakness in the other cohort. Furthermore, patients in the cohort with severe symptoms were older and had more frequent tumors in the thoracic spine, with a higher spinal canal occupancy ratio and spinal cord edema. The impact of the spinal canal ratio on neurological deficits was also demonstrated in previously published studies [2,21,24,25,26,27]. These all seem to be risk factors for developing more severe deficits or a delay in diagnosis [21]. We suppose the localization of tumors within the cervical and lumbar spine causes typical radicular symptoms and accelerates diagnosis and treatment [2,27]. The smaller spinal canal occupancy ratio, the younger age of the patients, and the less frequent spinal canal edema all indicate that these tumors were identified in an earlier stage of the disease. Moreover, imaging is much more frequently performed on the cervical and lumbar spine due to the higher rate of degenerative conditions in these localizations than imaging of the thoracic spine. We, therefore, encourage performing more imaging of the thoracic spine in patients with refractory back pain or when neurological findings are not correlated to findings in the more frequent images on the cervical and lumbar spine.

The results of this study indicate that resection of spinal meningiomas causing cord compression, even when causing only minor symptoms, should be treated in the early stage of the disease. Surgery in advanced stages may relieve symptoms in most cases, but the recovery would not be as effective as in patients with mild symptoms. A treatment strategy of wait and see, which is often propagated, seems to be less recommended than in other intradural tumors [22,28,29,30].

### 4.2. Effect of Neurological Outcome on Functionality and Quality of Life

The overall QoL of patients after resection of spinal meningiomas was relatively equivalent to the general population [20], with minor deficits in the subcategory of physical functionality. However, in this study, there were more elderly patients included, which also has an influence on physical functionality [20]. As expected, we observed an association between QoL and postoperative neurological outcomes. Patients who showed a favorable neurological outcome after tumor resection graded their life quality significantly higher than those patients who had incomplete recovery after surgery. This was significantly different in the subcategories “physical functionality” and “role limitations due to physical health”. Furthermore, we found a very strong negative correlation between the subcategory physical functionality and disability according to the NDI and ODI questionnaires, which are validated for disabled due to spinal diseases [9]. This indicates the influence of spinal diseases on the physical function subcategory of QoL [4].

Similar results on the influence of permanent disabilities due to spinal tumors were published before [2,10,21,31,32]. Other previously published studies highlight that supportive care of oncological patients who suffer from neurological symptoms reduces the rate of psychological disorders, pain, and anxiety [33,34,35].

### 4.3. Quality Indicators and Adverse Events

The role of QI has become more important in recent years [7]. Therefore, it is important to report the typical measured QI in different pathologies and medical procedures for future evaluations. With four adverse events cases (6.15%), two of them in one patient, the rate is similar to the previously published systematic review [23]. In this study, we found one patient who had a nosocomial urinary tract infection. This patient was 75 years old when admitted with paraplegia due to a meningioma at thoracic level 4–5 and in a bad general condition with a KPS of 50 due to cardiovascular conditions. After emergency surgery, she had to be admitted to an intensive care unit to further her condition due to further cardiac decompensation. Finally, she was dismissed to a nursing care facility after 18 days. Her neurological status indeed improved but remained very severe, with a McCormick scale of 4. All these conditions are risk factors for nosocomial infection and prolonged LOS [2,23].

In comparison to other studies, we did not identify epidural hemorrhages as adverse events [21], nor did we find any association between the localization of the tumor and complications [8,24]. Moreover, we noticed only one case of neurological deterioration after surgery in a patient with a preoperative McCormick scale 3. Therefore, we could not verify any significant association for neurological deterioration [21].

LOS was significantly lower in the cohort of patients with mild symptoms in comparison to those with more severe symptoms, and patients with favorable recovery also stayed in the hospital for shorter periods. Readmission within 90 days was reported once due to a CFS leakage, which had to be treated operatively. This patient had a favorable outcome.

Intraoperative neurophysiological monitoring (IOM) was utilized in all included cases. Its role is not yet verified for intradural tumors [2,26], but it may give the surgeon feedback on neurological conditions, especially when spinal cord manipulation is required [26,36]. We recommend using IOM for these procedures.

### 4.4. Minimal-Invasive Surgery

Previous publications also showed very good QoL after resection of spinal meningiomas and other benign intradural tumors [4,8,9,10]. However, these cases reported mostly on patients operated on via a bilateral laminectomy as the surgical approach. A unilateral surgical approach, as it is less invasive, was performed in most cases in the study. The main goal of minimal-invasive spine surgery (MISS) is to minimize the collateral damage both locally and systematically, without reducing the effectiveness of the main goal of the surgical procedure, safe and complete resection in this case [37,38]. The role and efficacy of minimal-invasive spine surgery were shown in several studies in degenerative spinal conditions, showing similar effects on decompression of the spinal canal, with quicker recovery, less pain, and shorter LOS [13,39]. These results advocate that more extensive approaches are not required for the resection of spinal meningiomas [40]. Figure 7 presents an illustrative case of minimal-invasive resection of a SM.

Previous publications also showed that these less invasive approaches were as effective as more invasive approaches to achieve GTR of spinal meningioma without additional side effects. GTR was achieved in 93.84% of all cases included in this study. Moreover, previous publications comparing both approaches show significantly less blood loss and lower LOS [11]. More invasive surgical approaches do not seem to be required in most cases, even in ventral-located and calcified meningiomas [11,28,36,41] and other intradural tumors [42]. Furthermore, minimal-invasive unilateral approaches seem to play a role in preventing CSF leakage, one of the more frequent possible complications after surgery on intradural pathologies [21,23,43,44]. In this study, we noticed one case of postoperative CSF leakage (1.54%). The enhanced recovery after minimal-invasive unilateral approaches may influence postoperative QoL and QI.

On the other hand, more extensive approaches, including facetectomy, are apparently required in order to achieve GTR on dumbbell tumors [45,46]. One patient in this study had to be operated on twice due to a dumbbell meningioma in the cervical spine. In the first surgery, only subtotal resection (STR) was achieved via a hemilaminectomy, and she had to undergo one more tumor resection due to progression after 5 years, again only achieving STR via hemilaminectomy.

In addition to surgical resection of spinal meningiomas, some studies showed efficacy in treating benign intra-spinal, intradural tumors, including meningiomas, with stereotactic radiosurgery (SRS) [47,48]. Although both studies report pain relief in treated patients, they did not show significant improvement in neurological deficits and mainly demonstrated growth control of the tumors. These results seem to be worse than the results of microsurgical resection [10]. According to the European treatment guidelines, SRS should be considered as a second-line treatment option, for example, in the case of recurrence of residual tumors [15].

### 4.5. Limitations

The study’s main limitation is the retrospective nature of the analysis. Hence, some of the analyzed scores were derived from medical reports with inherent limitations. Moreover, only 60% of the patients were available to fill out the questionnaire. The different time interval between surgery and recirculation of the questionnaires may also impact their validity. In addition, other medical conditions, such as cardiovascular diseases and neurological and psychological disorders, may also affect the results of the questions. A further limitation is the lack of comparison between operative and non-operative cases.

## 5. Conclusions

Minimal-invasive resection of spinal meningiomas is safe and effective, and patients experience very good outcomes after surgery. Neurological and functional outcomes and quality of life highly depend on preoperative findings. The results of our study recommend resection of spinal meningiomas in an early stage of the disease when patients are asymptomatic or have only mild symptoms, especially in the case of cord compression. We therefore recommend surgical resection of all spinal meningiomas with cord or nerve root compression. In the case of small meningiomas, without compression of any neural structures, we recommend careful surveillance with MRI, but preventive surgical resection may also be considered, as surgical results are very good and durable in most cases. Furthermore, we recommend performing MRI images of the whole spine in cases of persistent back or neck pain.

## Figures and Tables

**Figure 1 cancers-16-02336-f001:**
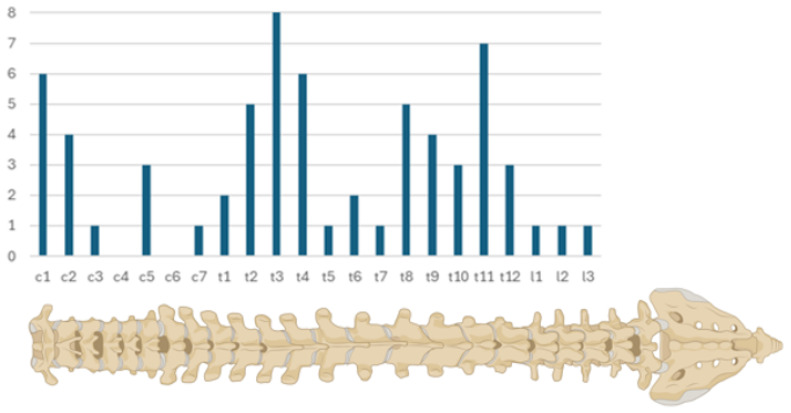
The cranio-caudal distribution of meningiomas within the spinal canal demonstrates notable peaks at the junction areas C1-C2, T1-T4, and T8-T12.

**Figure 2 cancers-16-02336-f002:**
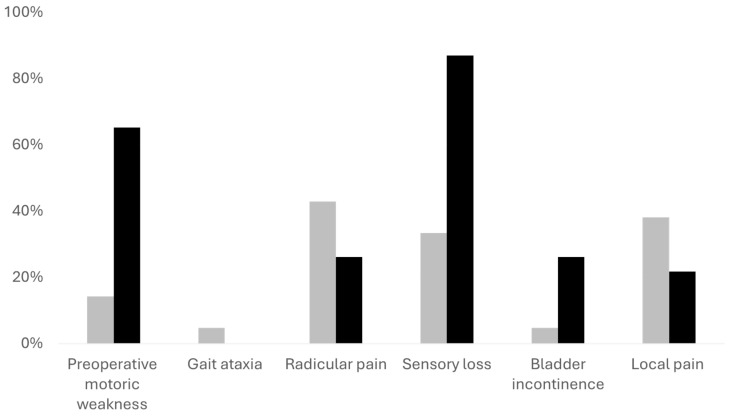
Overview of preoperative neurological symptoms. The gray bars represent patients with mild preoperative neurological symptoms, black bars represent patients with more severe neurological symptoms. The Y axis demonstrates the percentage of patients with each symptom in each cohort. Motor weakness, sensory deficits, and bladder dysfunction were significantly more prevalent in the cohort with severe symptoms (all *p* < 0.001).

**Figure 3 cancers-16-02336-f003:**
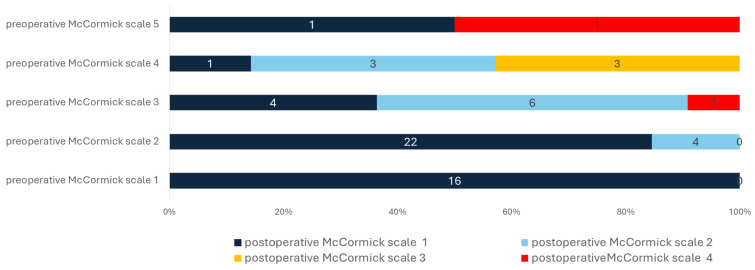
Relationship between preoperative and postoperative McCormick scale. All 16 patients with preoperative McCormick scale 1 had the same scale after surgery. Patients with preoperative McCormick scale 2 also had favorable outcomes with McCormick scale 1 in most cases (*n* = 22, 84.62%). On the other hand, patients with a McCormick scale of 3 to 5 had much lower odds of favorable outcomes after surgery (*p* < 0.001). See text for further information.

**Figure 4 cancers-16-02336-f004:**
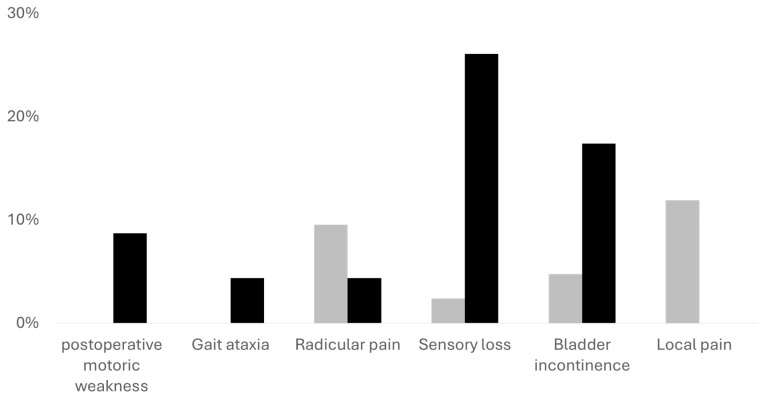
Remaining neurological symptoms after tumor resection. The gray bars represent patients with mild preoperative neurological symptoms, black bars represent patients with more severe neurological symptoms. The Y axis demonstrates the percentage of patients with each symptom in each cohort. Sensory deficits were significantly higher in the cohort with severe symptoms (*p* = 0.006). All other symptoms show no statistical significance.

**Figure 5 cancers-16-02336-f005:**
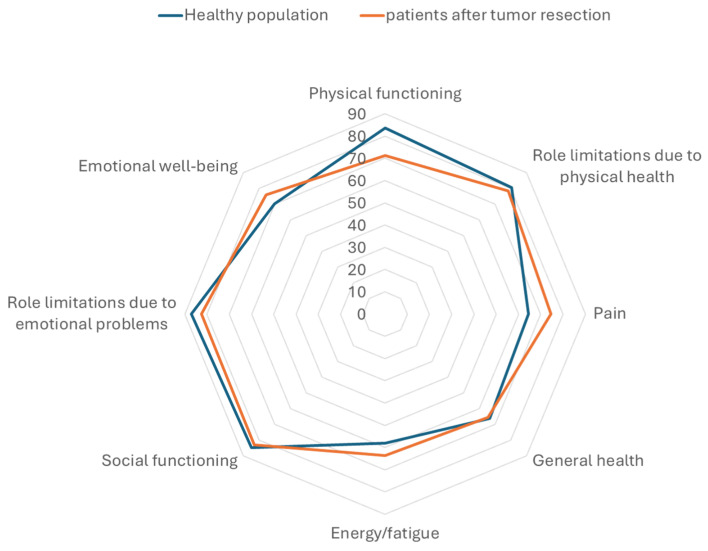
Comparison of quality of life according to the SF-36 questionnaire between the general population (blue) and patients included in this study (orange). Notice the relatively comparable results in the subcategories of the questionnaire, except for the slight difference in physical function and pain.

**Figure 6 cancers-16-02336-f006:**
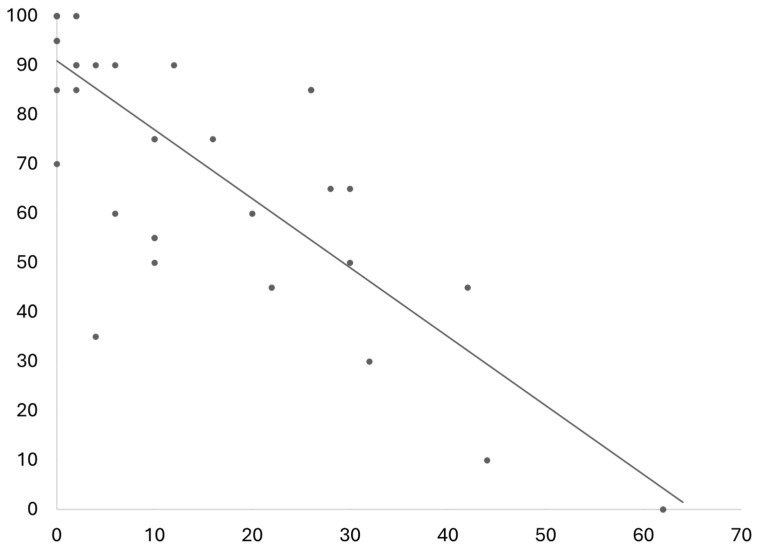
The relationship between disability scores ODI and NDI—on the Y axis—and the subcategory physical function in the SF-36 questionnaire—on the X axis—shows a significant negative correlation (r(36) = −0.83, *p* < 0.001).

**Figure 7 cancers-16-02336-f007:**
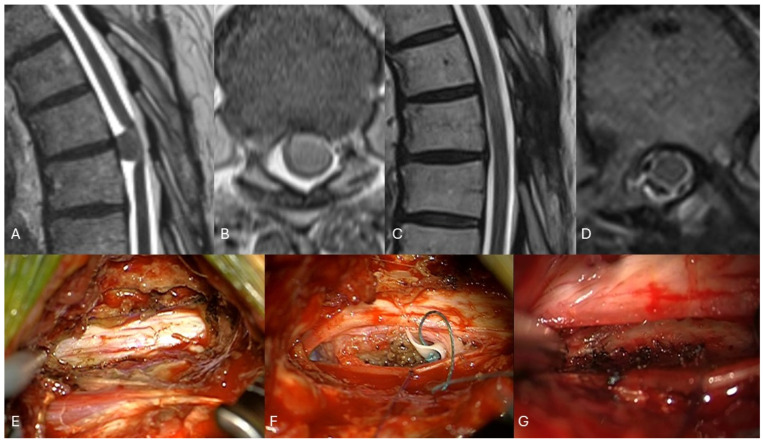
Case example of a 56-year-old male patient with a ventrally located meningioma in the thoracic spine. The patient was admitted with progressive ataxia and hemiparesis. A gross total resection was achieved via a left-sided posterolateral approach, and neurological deficits recovered quickly. Three months after surgery, no deficits were detected in the neurological examination. (**A**): preoperative T2 weighted sagittal MRI image showing the space-occupying lesion with compression of the spinal cord. (**B**): preoperative T2 weighted axial MRI image with compression of the spinal cord to the left. (**C**): postoperative T2 weighted sagittal MRI image demonstrating the enfolded spinal cord after tumor resection. (**D**): postoperative T2 weighted axial image demonstrating the enfolded spinal cord and the left-sided hemilaminectomy. (**E**): intraoperative image after partial removal of the left lamina showing the dura mater. (**F**): the dura was opened just dorsally to the nerve root in order to reach the ventral part of the spinal canal, the dentate ligament was cut, and the tumor was debulked using an ultrasonic aspirator (CUSA). (**G**): at the end of the procedure, the dura attachments were coagulated to achieve a Simpson grade 2 resection.

**Table 1 cancers-16-02336-t001:** Patient characteristics (SD: standard deviation).

Variable	Mild Symptoms (*n* = 42)	Severe Symptoms (*n* = 23)	*p*-Value
Age (years, mean ± SD)	55.29 (±13.52)	64.09 (±13.62)	0.015
Female patients (*n*, %)	28 (90.48%)	21 (91.30%)	
Karnofsky performance scale preoperative (median, IQR)	80 (70–80)	70 (60–70)	<0.001
Other spine procedures in the past (*n*, %)	1 (2.38%)	2 (8.70%)	0.284
Other neurological diseases (*n*, %)	2 (4.76%)	2 (8.70%)	0.603
Psychological disorders (*n*, %)	1 (2.38%)	1 (4.35%)	>0.99
Other meningiomas in different localization (*n*, %)	4 (9.52%)	3 (13.04%)	0.691
Preoperative McCormick scale (median, IQR)	2 (1–2)	3 (3–4)	<0.001
Preoperative McCormick scale 1 (*n*, %)	16 (38.1%)	0	<0.001
Preoperative McCormick scale 2 (*n*, %)	26 (61.3%)	0	<0.001
Preoperative McCormick scale 3 (*n*, %)	0	15 (65.22%)	<0.001
Preoperative McCormick scale 4 (*n*, %)	0	6 (26.09%)	<0.001
Preoperative McCormick scale 5 (*n*, %)	0	2 (8.70%)	0.114
Motor weakness (*n*, %)	6 (14.29%)	15 (65.22%)	<0.001
Gait ataxia (*n*, %)	2 (4.76%)	16 (69.57%)	<0.001
Radicular pain (*n*, %)	18 (42.86%)	6 (26.09%)	0.282
Sensory deficit (*n*, %)	14 (33.33%)	20 (86.96%)	<0.001
Bladder incontinence (*n*, %)	2 (4.76%)	6 (26.09%)	0.016
Local pain (*n*, %)	16 (38.10%)	5 (21.74%)	0.275
Duration of Symptoms (*n*, %)			0.093
More than 6 months (*n*, %)	12 (28.57%)	8 (34.78%)	0.780
Less than 6 months (*n*, %)	19 (45.24%)	14 (60.87%)	0.302
Unknown (*n*, %)	11 (26.19%)	1 (4.35%)	0.043
Primary case (*n,* %)	38 (90.48%)	23 (100%)	0.290
Recurrent tumor (*n*, %)	4 (9.52%)	0	
Localization within the spinal canal			0.087
Cervical spine (*n*, %)	14 (33.33%)	3 (13.04%)	0.087
Thoracic spine (*n*, %)	25 (59.52%)	20 (86.96%)	0.026
Lumbar spine (*n*, %)	3 (7.14%)	0	0.546
Tumor volume (ml, mean, SD)	1.13 (±0.79)	1.36 (±0.95)	0.294
Tumor/ spinal canal ratio (%, mean, SD)	47.25% (±17.77%)	59.73 (±21.07%)	0.016
Compression of spinal cord (*n*, %)	36 (85.71%)	23 (100%)	0.082
Spinal cord edema (*n*, %)	5 (11.90%)	8 (34.78%)	0.049

**Table 2 cancers-16-02336-t002:** Surgical data and postoperative outcome (SD: standard deviation).

Variable	Mild Symptoms (*n* = 42)	Severe Symptoms (*n* = 23)	*p*-Value
Duration of surgery (min, mean, SD)	238.34 (±111.77)	231.13 (±68.51)	0.785
Uni-lateral approach (*n*, %)	37 (88.1%)	22 (95.65%)	0.411
Bilateral approach (*n*, %)	5 (11.9%)	1 (4.35%)	0.411
Extent of resection			
Simpson grade 2 (*n*, %)	40 (95.24%)	22 (95.65%)	>0.99
Simpson grade 3 (*n*, %)	2 (4.76%)	1 (4.35%)	
Estimated blood loss (ml, mean, SD)	236.72 (±315.70)	356.13 (±384)	0.191
Length of hospital stay (days, mean, SD)	7.07 (±2.4)	10.04 (±5.36)	0.003
Adverse events (*n*, %)	2 (4.76%)	2 (8.7%)	>0.99
CSF leak (*n*, %)	1 (2.38%)	0	>0.99
Pulmonary embolism (*n*, %)	1 (2.38%)	0	>0.99
Cardial decompensation (*n*, %)	0	1 (4.35%)	0.354
Urinary tract infection	0	1 (4.35%)	0.354
Histology WHO 1 (*n*, %)	42 (100%)	23 (100%)	>0.99
Karnofsky performance scale postoperative (Median, IQR)	90 (90–100)	80 (70–90)	0.004
Postoperative McCormick scale (Median, IQR)	1 (1–1)	2 (1–2)	0.001
Postoperative McCormick scale 1 (*n*, %)	38 (90.48%)	9 (39.13%)	<0.001
Postoperative McCormick scale 2 (*n*, %)	4 (9.52%)	10 (43.48%)	0.003
Postoperative McCormick scale 3 (*n*, %)	0	3 (13.04%)	0.037
Postoperative McCormick scale 4 (*n*, %)	0	1 (4.35%)	0.354
Postoperative McCormick scale 5 (*n*, %)	0	0	>0.99
Postoperative motoric weakness (*n*, %)	0	2 (8.70%)	0.122
Gait ataxia (*n*, %)	1 (2.38%)	1 (4.35%)	>0.99
Radicular pain (*n*, %)	4 (9.52%)	1 (4.35%)	0.649
Sensory loss (*n*, %)	1 (2.38%)	6 (26.09%)	0.006
Bladder incontinence (*n*, %)	2 (4.76%)	4 (17.39%)	0.174
Local pain (*n*, %)	5 (11.90%)	0	0.152
Readmission in 90 days (*n*, %)	1 (2.38%)	0	>0.99
Re-Surgery in 90 days (*n*, %)	1 (2.38%)	0	>0.99
Tumor recurrence (*n*, %)	3 (7.14%)	0	0.547
Progression-free survival (years, mean, SD)	7.45 (±3.95)	6 (±3.40)	0.121

**Table 3 cancers-16-02336-t003:** Preoperative characteristics of patients who completed the distributed questionnaires.

Variable	Favorable Neurological Outcome (*n* = 27)	Incomplete Neurological Outcome (*n* = 11)	*p*-Value
Age at time of surgery (years, mean ± SD)	56.93 ± 12.62	64.64 ± 12.04	0.092
Age at present (years, mean ± SD)	63.15 ± 12.77	68.73 ± 12.06	0.223
Female patients (*n*, %)	25 (93%)	10 (91%)	>0.99
Karnofsky performance scale preoperative (median, IQR)	80 (70–80)	70 (70)	<0.001
Other spine procedures in the past (*n*, %)	3 (11%)	2 (18%)	0.615
Other neurological diseases (*n*, %)	2	1 (9%)	>0.99
Psychological disorders (*n*, %)	0	1 (9%)	0.290
Other meningiomas (*n*, %)	1	2	0.196
McCormick scale (median, IQR)			
McCormick scale 1 (*n*, %)	6	0	0.154
McCormick scale 2 (*n*, %)	16	1	0.010
McCormick scale 3 (*n*, %)	3	6	0.009
McCormick scale 4 (*n*, %)	1	4	0.019
McCormick scale 5 (*n*, %)	0	0	>0.99
McCormick scale 1–2 (*n*, %)	22	1	<0.001
Preoperative motoric weakness (*n*, %)	6 (22%)	5 (45%)	0.238
Gait ataxia (*n*, %)	0	10 (91%)	<0.001
Radicular pain (*n*, %)	13 (48%)	3 (27%)	0.296
Sensory loss (*n*, %)	10 (37%)	9 (32%)	0.029
Bladder incontinence (*n*, %)	1 (4%)	2 (18%)	0.196
Local pain (*n*, %)	11 (41%)	1 (9%)	0.121
Duration of Symptoms (*n*, %)			
More than 6 months (*n*, %)	5	5	0.116
Less than 6 months (*n*, %)	17	5	0.471
Unknown (*n*, %)	4	1	>0.99
Primary case (*n*, %)	24	11 (100%)	0.542
Recurrent tumor (*n*, %)	3	0	0.542
Localization with spinal canal			
Cervical spine (*n*, %)	7	1	0.395
Thoracic spine (*n*, %)	17	10	0.124
Lumbar spine (*n*, %)	3	0	0.542
Tumor volume (ml, mean, SD)	1.17 ± 0.81	1.26 ± 1.10	0.787
Tumor/ spinal canal ratio (%, mean, SD)	52 ± 17%	59% ± 22%	0.468
Compression of spinal cord (*n*, %)	25 (93%)	11 (100%)	>0.99
Spinal cord edema (*n*, %)	1 (4%)	4 (36%)	0.019

**Table 4 cancers-16-02336-t004:** Surgical data and neurological outcomes of the patients who completed the questionnaires.

Variable	Favorable Neurological Outcome (*n* = 27)	Incomplete Neurological Outcome (*n* = 11)	*p*-Value
Duration of surgery (min, mean, SD)	211.48 ± 95.87	236.09 ± 88.07	0.468
Unilateral approach (*n*, %)	26	11 (11%)	>0.99
Bilateral approach (*n*, %)	1	0	>0.99
Extent of resection			
Simpson grade 2 (*n*, %)	24 (89%)	10 (91%)	>0.99
Simpson grade 3 (*n*, %)	3 (11%)	1 (9%)	>0.99
Estimated blood loss (ml, mean, SD)	260.78 ± 382.88	203.73 ± 218.18	0.647
Length of hospital stay (days, mean, SD)	6.85 ± 1.26	9 ± 4.29	0.022
Adverse events (*n*, %)	0	0	
Histology WHO 1 (*n*, %)	27	11	>0.99
Karnofsky scale postoperative (Median, IQR)	100 (90–100)	80 (70–90)	<0.001
McCormick scale (Median, IQR)			
McCormick scale 1 (*n*, %)	27	0	<0.001
McCormick scale 2 (*n*, %)	0	9	<0.001
McCormick scale 3 (*n*, %)	0	2 (18%)	0.078
Postoperative motoric weakness (*n*, %)	0	1 (9%)	0.290
Gait ataxia (*n*, %)	0	1 (9%)	0.290
Radicular pain (*n*, %)	2 (7%)	2 (18%)	0.564
Sensory loss (*n*, %)	0	3 (27%)	0.020
Bladder incontinence (*n*, %)	0	0	>0.99
Local pain (*n*, %)	1 (4%)	1 (9%)	0.501
Readmission in 90 days (*n*, %)	0	0	>0.99
Re-Surgery in 90 days (*n*, %)	0	0	>0.99
Tumor recurrence (*n*, %)	2 (7%)	0	>0.99
Progression-free survival (years, mean, SD)	6.78 ± 4.14	4.09 ± 1.64	0.046

**Table 5 cancers-16-02336-t005:** Postoperative disabilities and quality of life.

Variable	Favorable Neurological Outcome (*n* = 27)	Incomplete Neurological Outcome (*n* = 11)	*p*-Value
ODI/NDI (mean, SD)	9.26% ± 11.60%	23.27% ± 24.10%	0.020
Physical functioning (mean, SD)	79.26 ± 20.03	51.82 ± 37.97	0.019
Role limitations due to physical health (mean, SD)	86.11 ± 27.15	59.09 ± 45.10	0.029
Role limitations due to emotional problems (mean, SD)	85.19 ± 29.72	75.75 ± 42.41	0.604
Energy/fatigue (mean, SD)	66.11 ± 21.81	57.27 ± 23.06	0.272
Emotional well-being (mean, SD)	78.48 ± 17.83	69.09 ± 22.42	0.180
Social functioning (mean, SD)	87.04 ± 18.50	73.86 ± 29.82	0.106
Pain (mean, SD)	80.19 ± 24.35	60.68 ± 32.66	0.050
General health (mean, SD)	68.33 ± 21.75	58.64 ± 22.59	0.226
Health change (mean, SD)	65.74 ± 25.14	61.36 ± 23.35	0.623

## Data Availability

Study data can be obtained by contacting the corresponding author at schwakem@uni-muenster.de.

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
