# Peer review of "Timing of Resection of Spinal Meningiomas and Its Influence on Quality of Life and Treatment"

_cancers, 2024, doi:10.3390/cancers16132336_

Round 1
Reviewer 1 Report
Comments and Suggestions for Authors
The authors investigate and show the characteristics of menigioma in spine.
The study is so interesting, however I have some concerns to discuss.
-What is the novelty of the current study?
-Is inflammation involved in the prognosis of the meningioma? Please discuss referring the following article.
Characterizing inflammatory markers in highly aggressive soft tissue sarcomas. Medicine (Baltimore). 2022;101(39):e30688. doi:10.1097/MD.0000000000030688
Author Response
The authors investigate and show the characteristics of menigioma in spine.
The study is so interesting, however I have some concerns to discuss.
-What is the novelty of the current study?
This study represents the first examination of quality of life (QoL) outcomes following the resection of spinal meningiomas using a minimally invasive approach. Additionally, it introduces the inaugural comparative analysis between the resection of meningiomas presenting with severe symptoms and those with mild symptoms, evaluating the impact on functional outcomes and QoL. The objective of this paper was to determine the justification for surgical intervention in cases with absent or mild symptoms.
-Is inflammation involved in the prognosis of the meningioma? Please discuss referring the following article.
Characterizing inflammatory markers in highly aggressive soft tissue sarcomas. Medicine (Baltimore). 2022;101(39):e30688. doi:10.1097/MD.0000000000030688
To date, there is no evidence to suggest that inflammation markers play a significant role in spinal meningiomas. While cranial meningiomas, especially secretory meningiomas and atypical variants, can lead to cerebral edema which can be treated with dexamethasone; such data are lacking for spinal meningiomas.
Reviewer 2 Report
Comments and Suggestions for Authors
I commend the authors for their commendable research efforts showcased in the manuscript titled "Timing of Resection of Spinal Meningiomas and Its Influence on Quality of Life and Treatment." This study aims to explore the optimal timing of resection for spinal meningiomas and its influence on quality of life (QoL) and quality indicators (QI).
The manuscript achieves a commendable level of clarity and readability, with a well-crafted introduction, a thorough presentation of the materials and methods, well-presented results, and a robust discussion. The inclusion of high-quality figures and tables further enhances the overall comprehensibility of the manuscript. An MRI depicting a typical case would be appreciated by the readers.
However, a major weakness in the study requires attention. Specifically, the conclusion section is rather scarce and represents the weakest part of the manuscript. The authors are encouraged to state clearly the uniqueness and importance of their meticulously crafted research. According to their results, what workflow would they recommend to their peers for managing patients with initial symptoms? Please elaborate.
In conclusion, while the manuscript provides valuable insights into current diagnostic and management strategies for spinal meningiomas, addressing the highlighted weakness through revisions will significantly enhance its impact and suitability for publication.
Author Response
I commend the authors for their commendable research efforts showcased in the manuscript titled "Timing of Resection of Spinal Meningiomas and Its Influence on Quality of Life and Treatment." This study aims to explore the optimal timing of resection for spinal meningiomas and its influence on quality of life (QoL) and quality indicators (QI).
The manuscript achieves a commendable level of clarity and readability, with a well-crafted introduction, a thorough presentation of the materials and methods, well-presented results, and a robust discussion. The inclusion of high-quality figures and tables further enhances the overall comprehensibility of the manuscript. An MRI depicting a typical case would be appreciated by the readers.
We thank the reviewer for the kind words made about our work and thank for the suggestion to add an illustrative case with MRI images. This has been added to the manuscript.
However, a major weakness in the study requires attention. Specifically, the conclusion section is rather scarce and represents the weakest part of the manuscript. The authors are encouraged to state clearly the uniqueness and importance of their meticulously crafted research. According to their results, what workflow would they recommend to their peers for managing patients with initial symptoms? Please elaborate.
We reformatted the conclusion part of the paper and added a recommended workflow. We recommend surgical resection of all spinal meningiomas with cord or nerve root compression. In the case of small meningiomas, without compression of any neural structures, we recommend careful surveillance with MRI, but preventive surgical resection may also be considered, as surgical results are very good and durable in most cases.
In conclusion, while the manuscript provides valuable insights into current diagnostic and management strategies for spinal meningiomas, addressing the highlighted weakness through revisions will significantly enhance its impact and suitability for publication.
Reviewer 3 Report
Comments and Suggestions for Authors
Dear Editor,
the main objective of the present study is to optimize the timing of surgery, for spinal meningioma. Stratifying the patients with (McCormick scale and evaluating the quality of life by questionnaires the authors conclude that Early surgical intervention before the development of severe neurological deficits, should be considered as it is associated with a favorable neurological outcome and quality of life. These attempted results based on collecting data need to be improved using tumor-grade and necrotic area if the size is the same in the two groups of study
please add information about radiotherapy treatment pharmacological treatment please add some histological evaluation of tumor grade
radiological imaging before and after surgery
please better document your population study not only statistical analysis for this journal
Comments on the Quality of English Language
The English is corrected some misspelling errors in the text
Author Response
The main objective of the present study is to optimize the timing of surgery, for spinal meningioma. Stratifying the patients with (McCormick scale and evaluating the quality of life by questionnaires the authors conclude that Early surgical intervention before the development of severe neurological deficits, should be considered as it is associated with a favorable neurological outcome and quality of life.
These attempted results based on collecting data need to be improved using tumor-grade and necrotic area if the size is the same in the two groups of study
Tumor grading according to the WHO classification is detailed in Table 1, with all patients in this study presenting with WHO grade 1 tumors. Grade 2 and 3 meningiomas are very rare in the spine. Table 1 also includes the tumor volume and the spinal canal occupancy ratio. Although tumor volume was comparable in both groups, the occupancy ratio was higher in the cohort with more severe neurological symptoms.
Please add information about radiotherapy treatment pharmacological treatment please add some histological evaluation of tumor grade
We added information on radiotherapy treatments. So far there is no specific pharmacological treatment for spinal meningiomas available outside of clinical trials and none of the patients received any experimental pharmacological treatment.
Radiological imaging before and after surgery
In the methods part we added information on postoperative imaging. First postoperative images are performed three months after resection. Afterwards, imaging was repeated after one year, subsequently, surveillance intervals were doubled after each unremarkable MRI.
please better document your population study not only statistical analysis for this journal
We added more information to describe the study population.
Round 2
Reviewer 1 Report
Comments and Suggestions for Authors
The authors replied well. The manuscript is suitable for publication.
Reviewer 3 Report
Comments and Suggestions for Authors
Dear authors
In the replay version, you answered all my requirements
I agree to publication
Comments on the Quality of English Language
Misspelling errors through the text